Visualization and quantification of mimetic musculature via DiceCT

Dickinson Edwin edwin_dickinson@ncsu.edu 1
Atkinson Emily 1
Meza Antonio 1
Kolli Shruti 1
Deutsch Ashley R. 2
Burrows Anne M. 3 4
Hartstone-Rose Adam 1
1 Department of Biological Sciences, North Carolina State University , Raleigh , NC , United States of America
2 Department of Anthropology, University of Florida , Gainesville , FL , United States of America
3 Department of Physical Therapy, Duquesne University , Pittsburgh , PA , United States of America
4 Department of Anthropology, University of Pittsburgh , Pittsburgh , PA , United States of America
Wilson Laura
Electronic publication date: 2020 Jun 16
Publication date: 2020
Volume: 8
Electronic Location ID: e9343
Received 2020 Jan 8; Accepted 2020 May 21
Copyright: ©2020 Dickinson et al.
Copyright year: 2020
Copyright holder: Dickinson et al.
License: This is an open access article distributed under the terms of the Creative Commons Attribution License, which permits unrestricted use, distribution, reproduction and adaptation in any medium and for any purpose provided that it is properly attributed. For attribution, the original author(s), title, publication source (PeerJ) and either DOI or URL of the article must be cited.
License URL: https://creativecommons.org/licenses/by/4.0/

Keywords: Lemuroidea, Facial expression, Digital dissection, Muscle volume

Funding: National Science Foundation IOS-15-57125 BCS-14-40599 This work was funded by the National Science Foundation (IOS-15-57125 and BCS-14-40599). There was no additional external funding received for this study. The funders had no role in study design, data collection and analysis, decision to publish, or preparation of the manuscript.

==============================
The muscles of facial expression are of significant interest to studies of communicative behaviors. However, due to their small size and high integration with other facial tissues, the current literature is largely restricted to descriptions of the presence or absence of specific muscles. Using diffusible iodine-based contrast-enhanced computed tomography (DiceCT) to stain and digitally image the mimetic mask of Eulemur flavifrons (the blue-eyed black lemur), we demonstrate—for the first time—the ability to visualize these muscles in three-dimensional space and to measure their relative volumes. Comparing these data to earlier accounts of mimetic organization with the face of lemuroidea, we demonstrate several novel configurations within this taxon, particularly in the superior auriculolabialis and the posterior auricularis. We conclude that DiceCT facilitates the study these muscles in closer detail than has been previously possible, and offers significant potential for future studies of this anatomy.

Introduction

Mimetic muscles—or the muscles of facial expression—are associated with social communication in visually oriented species. They can be broadly classified into three separate groups based on their location and function (muscles surrounding the external ear; muscles of the superciliary region/orbital region; and muscles of the mid-face/oral region), and all are intimately association with the skin of the face. Additionally, the platysma—which originates postcranially—inserts around the mouth and cheeks and contributes to oral expression. Over the past century, numerous anatomical reports have sought to describe the gross anatomy of these muscles in primates (e.g., Murie & Mivart, 1869; Lightoller, 1925; Sullivan & Osgood, 1925; Lightoller, 1928; Huber, 1930; Lightoller, 1934; Shibata, 1959; Seiler, 1970; Swindler & Wood, 1973; Seiler, 1977; Pellatt, 1979; Burrows & Smith, 2003; Burrows et al., 2006; Burrows, 2008; Burrows, Waller & Parr, 2009; Diogo et al., 2009; Powell et al., 2018). More recently still, analyses of variation in mimetic muscle organization between dogs and wolves have shone light on intriguing differences in communication resulting from the process of domestication (Kaminski et al., 2019). However, to date, all studies of these muscles have been limited to qualitative descriptions regarding the presence, absence, and orientation of particular muscles. This likely reflects a confluence of factors, including their small size, superficial positioning within the face, and intimate integration with the skin—each of which render the mimetic musculature difficult to individually excise and analyze using traditional gross dissection techniques. Consequently, alternative techniques for the analyses of muscle volume and configuration are necessary to more comprehensively evaluate these muscles.

In recent years, new imaging modalities such as diffusible iodine-based contrast-enhanced computed tomography (DiceCT) have emerged as a means of visualizing and quantifying myological data in situ (Gignac et al., 2016). Typically, X-ray computed tomography is reserved for the visualization of mineralized materials such as bone; however, pretreatment of specimens with a chemical stain (such as phosphomolybdic acid or, as in DiceCT, iodine) capable of binding to carbohydrates in soft tissues (Li et al., 2015) makes it possible to apply this technique to the analysis of myology. Using this technique, muscle volumes can be visualized within CT stacks, segmented, and digitally reconstructed—permitting the three-dimensional visualization of their form, and quantification of morphometric data such as surface area and volume (e.g., Cox & Jeffery, 2011; Jeffery et al., 2011; Baverstock, Jeffery & Cobb, 2013; Holliday et al., 2013; Cox & Faulkes, 2014; Gignac & Kley, 2014). Indeed, recent advances have even permitted the reconstruction of individual fascicles within whole muscle volumes (Kupczik et al., 2015; Dickinson, Stark & Kupczik, 2018; Dickinson et al., 2019; Dickinson et al., 2020; Sullivan et al., 2019).

As this methodology facilitates the visualization of subdermal tissues in situ, DiceCT circumvents the obstacles that limit our ability to quantitatively assess the relative proportions of the mimetic musculature. In so doing, it becomes possible to compare three-dimensional data on the size and configuration of these muscles across species, potentially unlocking new insights into mimetic diversity across primates within the context of communicative behavioral repertoires. Within this pilot study we demonstrate the potential of this technique for quantification of these under-studied muscles within a strepsirrhine taxon, the blue-eyed black lemur (Eulemur flavifrons). As these muscles have been qualitatively described from several closely-related species of lemur (e.g., Murie & Mivart, 1869; Lightoller, 1934; Burrows & Smith, 2003), the use of this taxon will facilitate comparison of our observations to those of these preceding studies.

Materials & Methods

Specimen preparation

Our specimen consisted of an adult male blue-eyed black lemur (Eulemur flavifrons) sourced from the Duke Lemur Center (DLC6655m), that died of natural causes prior to the acquisition of the specimen. The cranial length of the specimen was 91 mm. Preparation of the specimen for scanning was conducting following the technique of ‘reverse dissection’ outlined by Burrows et al. (2019). The head was first disarticulated from the neck, and precise incisions were made in order to divide the face bilaterally. One half of the face was removed from its bony attachments, then exposed to the air in order to desiccate any remaining connective tissues. No visible muscle tissue was left on the head following removal of the facial mask.

The resulting face mask was fixed in 10% buffered formalin for 48 h to preserve the tissue during the staining procedure. The mask was then submerged in a 2.625% w/v solution of Lugol’s iodine (I2KI), with care taken to rest the specimen within a natural configuration, with all muscles below the surface of the solution. After two weeks, the face masks were removed from this solution and rinsed with distilled water. The mask was then wrapped in damp paper towels and placed in a sealed plastic bag for 24 h before scanning.

Scanning

In preparation for scanning, the face mask was mounted onto floral foam (a low-density material) and secured with wooden toothpicks to eliminate wrinkles and to limit movement of the mask during scanning. Using a Nikon XTH 225 ST micro-CT system housed at the Shared Materials Instrumentation Facility (SMIF) at Duke University, the mounted face mask was scanned at 130 kV and 171 mA with a 0.125 mm copper filter. The resultant images were reconstructed in 16bit, resulting in an isometric voxel size of 0.0509 mm and scan dimensions of 2,553 × 1,602 × 1,337 slices.

Segmentation

From the reconstructed image stack (2D slice in Fig. 1; see data availability section for URL to full 3D dataset), we identified and segmented fifteen mimetic muscles: the frontalis, occipitalis, platysma, orbicularis oris, orbicularis oculi, anterior auricularis, superior and inferior posterior auricularis, superior and inferior auriculolabialis, levator labii, mentalis, mandibuloauricularis, tragicus, and depressor helicis, using Amira 6.3 (Thermo Fisher Scientific). Due to the thin nature of each muscle portion, muscles were manually segmented in multiple anatomical planes to aid interpretation and ensure that no muscle tissue was disregarded. Three-dimensional volumes (constrained smoothing, intensity = 3) were produced of each labeled muscle, from which the volume of each muscle was calculated (Table 1).

Figure 1 2D slice from the contrast-enhanced image stack of the facemask of Eulemur flavifrons, with visible mimetic muscles segmented.

Muscle abbreviations as follows: F, frontalis; OCC, occipitalis; SAL, superior auriculolabialis; OOR, orbicularis oris; P, platysma.

Table 1 Volumetric measurements of key muscles of facial expression within Eulemur flavifrons.

Muscle Region	Muscle	Inferred muscle function inE. flavifrons	Volume (mm3)	Proportion of Total Mimetic Musculature (%)	
Muscles of the Superciliary/Orbital Region	Frontalis	Elevation of the brow	68.86	6.4%	
Occipitalis	Posterior movement of the scalp	39.87	3.7%	
Orbicularis Oculi	Closure of the eye	329.96	30.4%	
Muscles of the Midface/Mouth Region	Orbicularis Oris	Pursing of the lips	459.34	42.4%	
Superior Auriculolabialis	Elevation of the corner of the mouth	69.53	6.0%	
Inferior Auriculolabialis	Retraction of the corner of the mouth	59.94	5.9%	
	Mentalis	Protrusion of the lower lip	9.65	0.9%	
	Levator Labii	Elevation of the upper lip	7.03	0.6%	
Muscles Surrounding the external ear	Anterior Auricularis	Elevation of the ear	3.66	0.3%	
Superior Posterior Auricularis	Flattening/retraction of the ear	14.96	1.4%	
Inferior Posterior	Flattening/retraction of the ear	8.17	0.8%	
Tragicus	Retracts/opens the ear	5.48	0.5%	
Depressor Helicis	Anterior flattening of the ear	6.18	0.6%	
Mandibuloauricularis	Anterior flattening of the ear	5.46	0.5%	

Results

Contrast-enhanced computed tomography and digital reconstruction of the facial mask of E. flavifrons permitted both three-dimensional visualization and volumetric quantification of the mimetic musculature within this specimen. In total, 14 muscle volumes were quantified, totaling 1,086.21 mm3. A visual representation of these muscles in E. flavifrons, including a superimposition of these muscles relative to the skull, is presented in Fig. 2. Volumes were measured for all muscles barring the platysma, as the infero-caudal portion of this muscle originating from the clavicle was absent from our face mask, such that any volume would be incomplete. A full summary of the mimetic muscles and their volumes is presented in Table 1.

Figure 2 (A) Digital rendering of the mimetic muscles of Eulemur flavifrons, derived from a contrast-enhanced face mask. (B) superimposition of the mimetic muscles onto the bony facial skeleton.

Muscle abbreviations as follows: F, frontalis; OCC, occipitalis; OOC, orbicularis oculi; SAL, superior auriculolabialis; IAL, inferior auriculolabialis; AA, anterior auricularis; PA(S), posterior auricularis (superior); PA(I), posterior auricularis (inferior); DH, depressor helicis; MA, mandibuloauricularis; TR, tragicus; LL, levator labii; OOR, orbicularis oris; M, mentalis; P, platysma. Pink shading (*) indicates the aponeurotic sheet that connects the frontalis and occipitalis portions of the combined occipito-frontalis complex; as this aponeurotic sheet is non-muscular, it was not stained and not segmented.

In addition to this volumetric data, the three-dimensional visualization of these muscles also enables us to compare the configuration and organization of these muscles to previous qualitative accounts of the mimetic muscles within other species of Lemuroidea (Table 2). Several muscles closely accorded with earlier descriptions; the orbicularis oris, tragicus, anterior auricularis, platysma, and frontalis all display similar patterns of organization to existing descriptions of these muscles within Lemuroidea and Lorisoidea. Similarly, fibers from the inferior auriculolabialis arise from the superior border of platysma, following descriptions of this muscle within Otolemur by Burrows & Smith (2003) but contrasting to descriptions by Lightoller (1934). The latter noted a small gap between these muscles in Eulemur macaco and a hybrid black lemur.

Table 2 Description of the arrangement of mimetic muscles analyzed in E. flavifrons relative to previous studies of the facial muscles in Lemuroidea and select other primate taxa.

Muscle	Previous descriptions	Arrangement withinEulemur flavifrons	
Frontalis	In Lemur catta, this muscle covers both sides of the skull with the fibers being more developed posteriorly (Murie & Mivart, 1869). In Eulemur macaco, the frontalis is relatively small but interlaces with the orbicularis oculi, with fibers ending into the skin of the eyebrow (Lightoller, 1934); in Otolemur this muscle is large, attaching to the scalp	This muscle accords with previous descriptions.	
Occipitalis	In Otolemur, this muscle is described as travelling to the level of the superior concha, giving rise to the anterior auricularis/atollens aurem (Burrows & Smith, 2003). In Lemur catta, the muscle is broad with strongly developed fibers in its posterior portion, while in Galago crassicaudatus it is thin, though retains well-developed fibers (Murie & Mivart, 1869).	This muscle accords with previous descriptions.	
Orbicularis oculi	In all previous descriptions, this muscle is formed by a thin sheet of circular fibres surrounding the rima palpebrarum. In a hybrid lemur described by Lightoller (1934), several fine fibers continue anteriorly towards the nose. In Otolemur, its superior extent is limited to the superciliary region, but inferiorly the muscle extends almost to the region of the upper lip, where it gives rise to the superior auriculolabialis muscle (Burrows & Smith, 2003)	This muscle largely accords with previous descriptions, but did not give rise to the superior auriculolabialis as was described in Otolemur.	
Orbicularis oris	In Lemur catta the orbicularis oris is described as elongated and narrow (Murie & Mivart, 1869). Across Lemuroidea, the muscle is described as a primitive ring of muscle by Huber (1930) and as a thickened ribbon around the lips, coupled with a flat, sheet, like body which extends outwards by Lightoller (1934). In Otolemur, the muscle is described as dense and sphincter-like, being occasionally integrated with the inferior extent of the levator labii (Burrows & Smith, 2003).	This muscle accords with previous descriptions.	
Superior Auriculo-labialis	Within Lemuroidea this muscle is described as being tightly integrated with the inferior auriculolabialis, with fibers from both muscles running from the ear towards the upper lip (Lightoller, 1934). Significant differentiation between superior and inferior muscles are reported, however, within Otolemur by Burrows & Smith (2003).	The superior auriculolabialis in E. flavifrons largely resembles the description of this muscle in L. niger (now E. macoco). It did not arise from the orbicularis oculi as described in Otolemur.	
Inferior Auriculo-labialis	Huber (1930) notes in both Lemuroidea and Tarsius a broad connection of the inferior portion of the auriculolabialis with the platysma. This same association is noted by Burrows & Smith (2003) in Otolemur, who observe strong integration with the platysma, with inferior fibers arising from the superior border of the platysma muscle at about one-third of its length; and in Lemur catta by Lightoller (1934), who describe this muscle as following the same plane as the platysma, with the two muscles sharing closely united fibers.	The inferior auriculolabialis in E. flavifrons closely resembles previous descriptions of this muscle across several species within Lemuroidea and Lorisoidea.	
Mentalis	In Lemuroidea, the muscle is reported to vary in size from very small with a few oblique fibers (Burrows & Smith, 2003) to large and fan-shaped (Lightoller, 1934). It originates muscularly from fibers of the orbicularis oris and runs to the alveolar margin of the mandible.	The inferior auriculolabialis in E. flavifrons closely resembles previous descriptions of this muscle, but shares closer associated with the platysma than described in previous accounts.	
Levator Labii	Described in Otolemur as a single band in 2/3rds of specimens and as two-pronged in 1/3rd of specimens (Burrows & Smith, 2003). Five of the six two-pronged cases were observed in one taxon (O. crassicaudatus). It passes from the skin of the rostrum/midface to insert into skin superior to the upper lip.	This muscle appears to be single-bellied, as in the case of 2/3rds of specimens of Otolemur, ad otherwise accords with previous descriptions.	
Platysma	In Otolemur, the platysma is described as a broad, flat muscle with horizontal fibers that extends from the commissure of the mouth to the region posterior to the ear. Superiorly, it gives rise to the inferior auriculolabialis muscle (Burrows & Smith, 2003). The muscle is similarly described in Lemur catta by Lightoller (1934).	The inferior auriculolabialis in E. flavifrons closely resembles previous descriptions.	
Anterior Auricularis	The anterior auricularis exists as a number of parallel bands running from the concha within Lemuroidea (Ruge, 1885); No true M. auricularis anterior is identified within Lemur niger (now E. macaco), rather there is a large and powerful M. orbito-auricularis (Lightoller, 1934); In G. gorilla, the anterior auricularis is attached by a tendinous plate to the frontalis (Huber, 1930); In Macaca mulatta, an anterior auricularis is imperfectly separated from the frontalis, inserting into fascia near the superoanterior portion of cartilaginous pinna (Burrows, Waller & Parr, 2009).	The arrangement of this muscle in E. flavifrons most closely accords to the description by Burrows, Waller & Parr (2009) in M. mulatta, being partially associated with the posterior border of the frontalis and lying towards the superoanterior border of the ear.	
Posterior Auricularis	In Lemuroidea, the M. auricularis posterior consists of three muscular bands, two inserting into the posterior surface of the auricular cartilage and the third into the lower pole of the concha (Lightoller, 1934); In M. mulatta, this muscle consists of two slips of relatively equal size that attach into fascia near posterior region of the cartilage of the pinna (Burrows, Waller & Parr, 2009)	In E. flavifrons, this muscle consists of two bellies as described by Burrows, Waller & Parr (2009); however, these were not of equal size. Rather, the superior belly possesses a volume almost twice as large as the inferior.	
Tragicus	Limited descriptions exist for this muscle within primates; in M. mulatta, it is described as a small muscle consisting of arcing fibers that run from the anterior edge of the helix of the pinna to the tragus (Burrows, Waller & Parr, 2009).	The arrangement of this muscle in E. flavifrons most closely accords to the description by Burrows, Waller & Parr (2009) in M. mulatta.	
Depressor Helicis	Attaches to the tragus and the anteroinferior concha in Otolemur (Burrows & Smith, 2003) and is similarly described within the genus Lemur by Ruge (1885). However, Burrows & Smith (2003) note that this muscle was absent in 35% of Otolemur specimens.	This muscle accords with previous descriptions.	
Mandibulo-auricularis	In Otolemur, this muscle consists of a short, wide set of vertically-oriented fibers, sitting immediately anterior to the depressor helicis. The muscle is similarly positioned across Lemuroidea (Lightoller, 1934).	This muscle accords with previous descriptions.	

A number of remaining muscles—most prominently the orbicularis oculi, superior auriculolabialis, and posterior auricularis—demonstrated significant variation from earlier descriptions of the mimetic muscles in adult strepsirrhines. Burrows & Smith (2003) report integration between fibers of the orbicularis oculi and superior auriculolabialis within Otolemur, with the o. oculi giving rise to the superior auriculolabialis. Within E. flavifrons, however, no interdigitation between fibers of these two muscles was observed when examining the muscles in digital space. We did, however, note significant integration between the superior auriculolabialis and posterior-inferior fibers of the frontalis muscle. Anterior fibers of the superior auriculolabialis both integrate with fibers from the posterior frontalis and, in some regions, overlap such that these fibers sit immediately deep to overlying fascicles from the frontalis. This integration is not reported within Otolemur (Burrows & Smith, 2003) nor Eulemur macaco and the hybrid black lemur described by Lightoller (1934).

Finally, we note variation in the configuration of the posterior auricularis from descriptions of this muscle by both (Lightoller, 1934) and Burrows & Smith (2003). Within Eulemur macaco and the hybrid black lemur, this muscle is described as comprising three bellies, while Burrows & Smith (2003) describes two distinct bellies of equal size in Otolemur. The organization of this muscle in E. flavifrons is similar to that of Otolemur in comprising two bellies; however, within our specimen the superior belly possesses a volume almost twice as large as the inferior belly (14.96 vs. 8.17 mm3).

Discussion

Digital reconstruction of the facial mask of E. flavifrons makes it possible to report, for the first time, quantitative data on the size of these highly-integrated muscles (Table 1). Our visualizations also enable us to compare the configuration and organization of these muscles to previous qualitative accounts of the mimetic muscles within other species of Lemuroidea. As described above, several muscles closely accorded with earlier descriptions. Additionally, a minor deviation from one earlier description was observed within the inferior auriculolabialis.

A number of remaining muscles demonstrated more significant variation from earlier descriptions. Burrows & Smith (2003) report integration between fibers of the orbicularis oculi and superior auriculolabialis within Otolemur, with the O. oculi giving rise to the superior auriculolabialis. Within E. flavifrons, however, no interdigitation between fibers of these two muscles was observed when examining the muscles in digital space. We did, however, note significant integration between the superior auriculolabialis and posterior-inferior fibers of the frontalis muscle. Anterior fibers of the superior auriculolabialis both integrate with fibers from the posterior frontalis and, in some regions, overlap such that these fibers sit immediately deep to overlying fascicles from the frontalis. This integration is not reported within Otolemur (Burrows & Smith, 2003) nor Eulemur macaco and the hybrid black lemur described by Lightoller (1934). Finally, we note variation in the configuration of the posterior auricularis from descriptions of this muscle by both Lightoller (1934) and Burrows & Smith (2003). Within Eulemur macaco and the hybrid black lemur, this muscle is described as comprising three bellies, while Burrows & Smith (2003) describes two distinct bellies of equal size in Otolemur. The organization of this muscle in E. flavifrons is similar to that of Otolemur in comprising two bellies; however, within our specimen the superior belly possesses a volume almost twice as large as the inferior belly (14.96 vs. 8.17 mm3).

By combining the reverse-dissection technique with a high-resolution DiceCT protocol, we successfully establish herein a novel method for the three-dimensional visualization and quantification of the mimetic musculature. This study provides further evidence of the value of contrast-enhanced tomographic techniques in permitting the quantification of skeletal muscle volumes, following earlier studies (e.g., Cox & Jeffery, 2011; Baverstock, Jeffery & Cobb, 2013; Lautenschlager, 2013; Cox & Faulkes, 2014; Dickinson, Stark & Kupczik, 2018). Indeed, this protocol enabled the quantification of muscles with volumes as small as 4–10 mm3. However, it should be noted that not all portions of the face could be visually resolved; specifically, the nasal/rostral region presented small muscles which were highly integrated with connective tissues in an inconsistent manner, such that anatomical boundaries between muscle bodies could not be confidently discerned, and individual fascicles could typically not be resolved. It is therefore possible that sub-region scanning of this anatomy at even higher (<20 µm) resolutions may be necessary to accurately quantify this musculature.

Though these initial results appear promising, several caveats to the present study should be noted. Firstly, the inherent intra-specific variation in the organization of the mimetic musculature (as demonstrated in humans; e.g., (Watanabe, 2016)) precludes confident interpretation of evolutionary trends from single-specimen samples. Additionally, it is important to consider the potential for volumetric modification of muscles during the preparation process. Indeed, excision, the dessication of extraneous tissues, and staining could all potentially impact the final volumes by inducing shrinkage of superficially exposed tissues. Therefore, while the current study applies this protocol to an isolated facemask, future studies of mimetic muscle organization may wish to explore the anatomy of these muscles in situ. Although this method would likely necessitate a longer staining time, the mimetic muscles can be clearly seen in some DiceCT specimens that have been prepared for analysis of other regions (e.g., the focal specimen from (Dickinson et al., 2020)). In situ quantification would also enable the visualization of mimetic muscle tissues in direct association with the underlying morphology of the face, as well as minimizing the potential for muscle deformation and shrinkage associated with the excision and staining process. Though this effect appeared relatively minimal for the muscles analyzed herein, the extreme borders of several muscles displayed a slight curvature interpreted to reflect an unnatural curling of the muscle tissue into itself. In situ analyses in which surrounding tissues can stabilize these muscles may therefore yield even more precise data on muscle organization within the face. This would in turn allow, for the first time, the three-dimensional visualization of mimetic muscles from their origins to their insertions.

Conclusions

The application of DiceCT permits both the visualization of the mimetic musculature in three-dimensional space and, for the first time, the quantification of muscle volumes for these small and highly-integrated tissues. These novel data further demonstrate the potential for contrast-enhanced tomographic techniques in enhancing our anatomical understanding of small and obscured structures. Future studies may apply an adapted protocol to visualize these tissues in situ within other taxa, and provide more spatial context for these muscles in relation to other tissues of the face.

We would like to thank the Editor, two anonymous reviewers, and Philip Cox for their comments during the review process, which have helped to clarify and improve this manuscript. We are grateful to Justin Gladman for assistance with scanning, and the Duke Lemur Center for providing the specimen. This work was performed in part at the Duke University Shared Materials Instrumentation Facility (SMIF), a member of the North Carolina Research Triangle Nanotechnology Network (RTNN). This is Duke Lemur Center Publication Number # 1458.

Additional Information and Declarations

Competing Interests

Author Contributions

Data Availability

The authors declare there are no competing interests.

Edwin Dickinson conceived and designed the experiments, performed the experiments, analyzed the data, prepared figures and/or tables, authored or reviewed drafts of the paper, and approved the final draft.

Emily Atkinson, Antonio Meza and Shruti Kolli performed the experiments, analyzed the data, prepared figures and/or tables, authored or reviewed drafts of the paper, and approved the final draft.

Ashley R. Deutsch performed the experiments, prepared figures and/or tables, and approved the final draft.

Anne M. Burrows conceived and designed the experiments, analyzed the data, authored or reviewed drafts of the paper, and approved the final draft.

Adam Hartstone-Rose conceived and designed the experiments, analyzed the data, prepared figures and/or tables, authored or reviewed drafts of the paper, and approved the final draft.

The following information was supplied regarding data availability:

The CT data is available at Morphosource: https://www.morphosource.org/Detail/SpecimenDetail/Show/specimen_id/29847.

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
