# Peer review of "Visualization and quantification of mimetic musculature via DiceCT"

_PeerJ, doi:10.7717/peerj.9343_

## Round 0.1 · original submission · Major Revisions

I have received detailed comments from three reviewers on your manuscript and all have commented positively that your study is clearly presented, well-written and would be a welcome addition to the literature. Two reviewers raised some more substantial suggestions for revision. Overall, these comments largely align and indicate that some areas of the manuscript, particularly the methods relating to factors affecting staining and preparation as well as basic reporting for how the CT data were handled, require additional detail and justification. I agree with the reviewers and I think these additions will improve the reproducibility of this study and will further encourage others to explore diceCT in their research via this workflow.

I also note, as have all three reviewers, that the raw CT stack is not currently available nor are any 3D outputted data. Please ensure that you adhere to the data policy of PeerJ and provide raw data when submitting your revision. I look forward to receiving your revised text.

·

Basic reporting

The manuscript is very well written in excellent English. The introduction is concise but provides sufficient background to the topic with all major previous literature cited. The results section is relevant to the stated aim of the paper.

It isn't stated that the diceCT scans or the 3D reconstructions are going to be shared. Will these be made available via one of the morphological repositories?

Experimental design

This manuscript reports a very interesting study into the facial musculature of a strepsirrhine primate species. It’s really nice to see diceCT being used to look at facial muscles as the focus is so frequently the masticatory musculature and the more superficial muscles tend to get overlooked. The methodology seems sound and well-reported. The subject of the study is well-suited to publication in PeerJ and I am always pleased to see descriptive anatomy papers – this is important information that needs to be made available to other researchers.

Validity of the findings

The findings of this study seem valid and the results have not been over-interpreted. However, I feel that at the moment, this manuscript is somewhat superficial and requires a great deal more detail before it is suitable for publication. I outline what I think is missing in the general comments section.

Additional comments

1. The results section needs greatly expanding in order to describe each muscle in detail. I realise that the method used, in which the face is removed from the bone before staining may make this a bit more difficult, but I’m sure there is something that could be said where each muscle runs from and to, the orientation of the fibres, how the muscle is positioned with respect to the other muscles under study, and some indication of its function.

2. Related to the above point, at the moment there is only one figure in the whole manuscript. I was wondering if there were any other figures that would help reveal the morphology of the muscles under study. For instance, are there any slices from the microCT stack that would show the arrangement of some of the muscles (it’s a little unusual to not have at least one CT slice in a diceCT paper), or that could highlight some of the detailed anatomical points you raise in the discussion? Also, given that you’re dealing with superficial musculature, I was wondering whether it might be possible to produce a figure similar to 1B but with the reconstructed mimetic muscles laid over the pre-dissected head? That last point is just a suggestion – it may not be as useful a figure as I think it is, and you may not have a pre-dissection photo of the specimen in the correct orientation.

3. You’ve given us the mass of each muscle, but this information is only useful if we know how big the specimen was. Can you please report the body mass and/or cranial length of the specimen.

4. In the discussion, where differences between the morphology reported here and that described in other strepsirrhines are highlighted, I think it would be good to mention the function of the muscles that are being compared. Ideally, you could then speculate on the impact of the anatomical differences that you are reporting. I suspect that the differences are quite minor so it will be difficult to determine the functional consequences, but by at least reporting the function of the muscles, the reader can begin to assess the importance of these differences.

Reviewer 2 ·

Basic reporting

The language is clear and unambiguous. Professional language is used throughout. I did not have a problem understanding the goals, scope, findings, or conclusions of the study.

I recommend drawing more widely from the literature. For example, the blue-eyed black lemur is referred to as a model taxon, but no references are given to demonstrate this. My primatology colleagues don’t necessarily agree that it is a model, so references would help to bolster this statement. Likewise, sourcing of interpretations for mimetic musculature from the literature should be more fleshed out.

Visual aids and data provided are sufficient and necessary. I could see an argument being made to show some 2D diceCT images for mimetic musculature, but I also recognize that such thin slips of muscle are unlikely to be visually informative to the reader. What the authors have chosen to present is clear and well delinated.

This study does not incorporate a hypothesis because it is an anatomical description and proof of concept.

Experimental design

The experimental design revolves largely around specimen preparation, staining, micro-CT scanning, and digital dissection. All are well articulated, and the study is repeatable.

I recommend elaborating on what standard approach was used to distinguish overlapping musculature (e.g., could fascicle orientation be distinguished clearly at 50-micron scale resolution?). If multiple reference guides were used when interpreting overlapping muscles, this would be helpful to the reader as well.

The study does not appear to be ethically flawed. The sample was sourced opportunistically, and the interpretative work for 3D rendering musculature is based on comparative analyses with closely related taxa.

Validity of the findings

The study is novel. The authors make a good argument for the difficulty in working with mimetic musculature as a result of how thin it is and its placement within the skin. DiceCT and other contrast-enhanced 3D visualization methods are well suited to finely parse thse muscles in 3D space. Utilizing reverse dissection and a “mask” approach for staining the tissues of interest was clever, allowing the authors to minimize iodine concentrations. This should have enabled them to develop nice 2D images on which the 3D reconstructions are based. I suspect others will mirror this approach in the future when undertaking diceCT work on restricted regions of interest.

The authors do not appear to provide the underlying data. These could be in the from of the CT-generated images stack and metadata, and/or in the form of 3D .STL/.PLY/.OBJ files for the skull and mimetic musculature, and/or in the form of fly-through videos that provide 2D anatomical references to the 3D models. PeerJ typically requires data sharing with publications, and I would encourage the authors to make their datasets available as one or more supplemental files.

Conclusions are well stated and speculation is constrained to the topic of variation and comparisons with other approaches for studying memetic musculature. It is a sober and useful examination of the method and the opportunities

Additional comments

Please find below a few line-by-line suggestions and comments that I anticipate will further strengthen the manuscript:

Line 27–28 : the authors should in the neck in their introduction about memetic musculature. They note the platysma later in the study (and, reasonably, why the muscle volume is not quantified), but at this stage it would be more complete to include the neck in addition to regions of the face as locations of mimetic musculature.

Line 37: I recommend removing the “/” between “directionality/presence/absence” and instead structuring the content as “presence, absence, and orientation”, which is easier to read and does not equate the value of presence/absence with directionality as is implied by the current construction

Line 39: change “them” to “mimetic musculature” for clarity

Line 44: capitalize the “X” in “x-ray”

Line 57: delete the first “the” and delete the only comma

Line 58: delete the hypthen before “the blue-eyed” and replace it with a comma

Line 59: change “within” to “from”

Line 61: the blue-eyed black lemur is not considered a model taxon by all practitioners—I recommend supporting this statement with references or, failing that, deleting the word “model”

Line 71: Please clarify if any muscle tissue was left behind on the head during reverse dissection.

Line 91: What standard approach as used to distinguish overlapping musculature. Was one or more reference guides used?

Line 98: delete the duplicate “as the”

Line 116: Add “We observed that” to the beginning of this sentence.

Line 118: change the comma to a period and replace “who note” with “The latter noted” (this serves to shorten an overly long sentence

Line 118: hybrid with what other species?

Line 120: I recall that prior work was only done on adults, but it would be useful to reiterate here that you are comparing your work with muscles described in adults. As you discuss variation, the potential difference in ontogenetic staging would seem like a relevant point.

Line 136: Formal writing in American English recommends that a period follows the abbreviation “vs.”

Line 148: ~ 30–35-micron resolution is often considered the threshold beyond which individual fascicles cannot be discerned reliably. I agree with the point made in line 148, but
it would also be helpful if the authors clarified whether they were segmenting by identifying fascicles themselves or bundles of fascicles, including what was interdigitating
between mimetic muscles. At 50-micron resolution, I suspect the authors are working with bundles of fascicles. This point may be important others deciding how scanning
resolutions for their diceCT projects.

Line 160: change the first comma to a period. Chance “and” to “This would in turn”

Line 161: I am not confident that it can state accurately that no one has visualized mimetic muscles from their origins to their insertions. Even gross dissection will allow for this. I suspect the authors mean in 3D, or digitally, or with a certain level of accuracy/precision. They should adjust this statement so that says when they specifically mean it to say.

Line 163: Change “3D” to “three-dimensional” — although the term three-dimensional has been used several times already in the manuscript, it has not been abbreviated previously.

Line 165: add a “for” in between “potential” and “contrast-enhacned”

Figure 1 Caption: Abbreviations appear to inconsistently use capital and lower cases and parentheticals. If the abbreviations used here are considered standards in the field, providing a reference would be helpful. If not, I would suggest standardizing the abbreviation scheme. This
will make it easier for the reader to go back and forth between the abbreviations in the figure and their definitions in the legend.

Figure 1: Please insert a scale bar

Reviewer 3 ·

Basic reporting

This paper is written clearly and is well structured overall. Please correct minor spelling/formatting issues commented in the attached PDF.
While the brevity of the paper is good, this should not come at the expense of reporting key aspects of the method to allow reproducibility, or further exploration of results – there are several important aspects of the paper that require elaboration.
I feel that in its current form this work does not meet the criteria for a ‘minimum unit of publication’, and requires additional data from the literature to give meaning and context to the results reported here (please see results and discussion comments).

Experimental design

The scope of literature covered establishes the knowledge gap to be filled, however some additional implications and justification should be provided (please see general comments).

Validity of the findings

Some additional underlying data should be provided (please see general comments regarding Figure 1).

Additional comments

General Comments:
1. Understanding of 3D facial muscle anatomy in lemurs is a worthy and very interesting goal, but application of these staining and visualisation methods to an intact cadaveric head, where the facial muscles lie in situ, would be much more appropriate and informative (stain times for the very superficial facial muscles would not increase much from those reported here). It is understandable that the data here may represent a convenient/opportunistic sample acquired through past research, but the authors should discuss this more plainly in the paper, as their apparent choice not to use an intact head seems an important and obvious oversight for a study of this kind. The aims of the study (identification of anatomical differences between taxa and quantification of muscle volumes) cannot be adequately met using only a single facial ‘mask’ specimen that has been excised from the skull and dried (!) before staining. Via the method used here, muscle volumes will be modified (especially in muscles with high relative surface area) and meaningful data on muscular attachments to bone or associated tissue are lost (please see comments on method). Because of these limitations, the authors need to use more circumspect language in their stated aims and discussion of their findings.

2. A key justification for this study is that facial muscles have never before been visualised in 3D using DiceCT – but this can be taken further. In the introduction, the authors should expand their rationale for and implications of understanding facial musculature configuration in lemurs – i.e. why look at facial muscles in lemurs in particular? Can this tell us anything e.g. about the evolution of facial communication in primates (obviously beyond the scope of the present study, but worth noting as a future avenue of inquiry)? This would help to better establish the relevance of the current work.

3. Figures: The superimposition of the muscles on the skull in the lower Figure 1 panel is a good inclusion and provides important context for the segmented volumes above. However, the stained CT image stack data with which such images were produced is necessary to demonstrate to the reader the efficacy of your stain protocol and ease with which the muscle could be discerned from the surrounding tissue. In the absence of an included complete image stack as supplementary material, please add at least a panel to this figure (or an additional figure) showing select representative slices from which your volumes were segmented. You may also consider including slices through the rostral region to demonstrate how the integration of tissues here made segmentation of muscles in this area impossible.

Results & Discussion:
1. Shrinkage. The authors state in the methods that the dissected mask is deliberately desiccated prior to fixation and staining (line 70) – for how long, at what temperature? These need to be reported. Drying the tissues out will reduce the muscle volumes and affect the results reported. While I understand how this drying would be necessary for manual (traditional) wet dissection to better distinguish muscle from connective tissues, it should not be necessary in this case as the iodine stain more readily binds to muscle than connective tissue. In the discussion the authors say the deformation and shrinkage of muscles ‘appeared relatively minimal’ (line 157), but as no baseline or progress scans/photos are provided there is no basis on which to make this assessment. I believe the extent of this limitation on the study is strongly understated in the present manuscript, and the nod to this is the discussion (lines 153-161) is insufficient. The stain concentration was indeed quite low (though not as low as reported, see my comments on the method) and the stain duration was short, which both act to reduce the effects of shrinkage. However, multiple authors have shown that even at low (~2%) I2KI concentrations, muscle volume shrinkage can exceed 20% (Bribiesca-Contreras, F., & Sellers, W. I. (2017). Three-dimensional visualisation of the internal anatomy of the sparrowhawk (Accipiter nisus) forelimb using contrast-enhanced micro-computed tomography. PeerJ, 5, e3039; Vickerton, P., Jarvis, J. and Jeffery, N. (2013), Concentration‐dependent specimen shrinkage in iodine‐enhanced microCT. J. Anat., 223: 185-193. doi:10.1111/joa.12068). Please address in more detail the possible effects of shrinkage on the outcome of this study.
2. Comparative context. Because muscle volumes form the primary quantitative component of this work, lack of context is a critical issue with the paper. Simply stating the volumes in isolation (especially considering shrinkage and ex situ issues identified above) has minimal scientific value and these data need to be presented with some sort of comparative perspective. My suggestion to rectify this would be to express these volumes as proportions of overall total mimetic muscle volume (though this assumes that all tissues have shrunk equally relative to one another and this should be discussed). Then, these proportions can be compared to any existing data on mimetic muscle volume in other primates (or indeed mammals) – please seek this out and incorporate these comparative data into the paper. The authors are probably aware of this recent paper providing image stacks of several DiceCT stained lemur heads on MorphoSource.org (Yapuncich GS, Kemp AD, Griffith DM, Gladman JT, Ehmke E, et al. (2019) A digital collection of rare and endangered lemurs and other primates from the Duke Lemur Center. PLOS ONE 14(11): e0219411. https://doi.org/10.1371/journal.pone.0219411), which can offer raw comparative data to this end (these would also provide additional qualitative comparisons which would strengthen the anatomical portion of the paper). If the authors are able to find older myological studies that may instead report mimetic muscle masses, these would also be appropriate and could be converted to volumes using standard mammalian muscle density values.

3. Intraspecific variation. The anatomical discrepancies found by the authors between theirs and existing work on closely related lemurs are quite minor. Facial communication in lemurs has not been demonstrated (yet!), so it is likely that facial muscles are under low selective pressure and therefore may vary intraspecifically to a large degree. Even in humans, where facial communication is extremely important, facial muscle presence/absence, arrangement and characteristics vary considerably (e.g. see Watanabe, K. (2016). Facial Muscles and Muscles of Mastication. In Bergman's Comprehensive Encyclopedia of Human Anatomic Variation. doi:10.1002/9781118430309.ch24). Whether the anatomical differences noted here are typical or not for Eulemur flavifrons cannot be stated based on a single specimen. Please elaborate on this in the discussion using more cautious language, addressing the ‘n=1’ sample size and strong possibility of wide intraspecific variation in these traits.

Method:
1. Missing details. More detail is needed in the methods to make them reproducible and align with emerging conventions in DiceCT literature:
• How many slices were in the final CT image stack?
• Did the authors do any resampling etc when dealing with a large (?) image stack in Amira? If so, please report this.
• How were the boundaries of muscle tissues discerned from surrounding soft tissues? (by eye is fine of course, but the authors should provide examples of their image slices so the reader can see the efficacy of the stain, and thereby the reliability of visual distinction between muscle and non-muscle)
• Details of software tools used, and any automated software processes. Was every slice manually segmented or was there inter-slice interpolation?
• Did the segmented muscles undergo any smoothing when generating the 3D meshes? This may affect the volumes calculated and should be reported if used.

2. Lugol’s concentration reporting issue. I believe the concentration of Lugol’s solution has been misreported and is slightly higher than stated (this is minor). The authors cite a paper (Burrows, A. M., Omstead, K. M., Deutsch, A. R., Gladman, J. T., & Hartstone-Rose, A. (2019). Reverse Dissection and DiceCT Reveal Otherwise Hidden Data in the Evolution of the Primate Face. Journal of Visualized Experiments, (143). doi:10.3791/58394) for the reverse dissection protocol, which I followed up as the stain concentration was not one I had come across before. The stain concentration reported in Burrows et al. 2019 may be incorrect (1.75g iodine plus 3.5g potassium iodide in 200mL water is 2.625% w/v I2KI, not 1.75% as reported). If this is the method used to mix the Lugol’s iodine for this study, please check this and report the correct concentration in the methods here.

Annotated reviews are not available for download in order to protect the identity of reviewers who chose to remain anonymous.

---

## Round 0.2 · Minor Revisions

Thank you for your detailed and thorough response to the comments raised by the reviewers. Both reviewers have concluded that the manuscript is much improved by these additions, and as such both are happy to recommend acceptance. I agree with the reviewer's thoughts and thank the authors for their careful revisions. Prior to acceptance, Reviewer #3 has requested that the authors provide some additional information about the image slice shown in Fig 1. This will be very quick to add, and once the authors have updated the text, I happy to recommend your manuscript moves to production.

·

Basic reporting

The language used throughout is good. Sufficient background has been provided and the study is self-contained with results relevant to hypotheses. I'm very pleased to see the new figure of a diceCT slice - I think it makes it very clear how difficult the reconstruction would have been in some cases. The new table is excellent - a very useful summary of the anatomy.

I think there is still a problem regarding the data sharing - I followed the link to Morphosource and I can find the author's project, but the diceCT stack doesn't seem to be there. It says 'This specimen has no media'.

Experimental design

I am satisfied that the experiment is well-designed and described in sufficient detail.

Validity of the findings

The results are well-described, particularly now that Table 2 has been included. The conclusions are justified.

Additional comments

I am happy that the authors have addressed all my comments and, as far as I can see, the comments of the other reviewers.

Reviewer 3 ·

Basic reporting

Thank you for providing the complete image stack data on Morphosource.
The addition of the colour coded 2D slice in Figure 1 is very helpful.
To assist the reader in interpreting this figure, please add some orientation labels to the figure itself or to the caption (e.g. superior at top, superficial at right etc.), and please add in the caption which plane the section is in (coronal, sagittal etc.).

Experimental design

The methodological clarifications and inclusion of some more cautious language have addressed all the concerns I had.

Validity of the findings

No comment

Additional comments

The additional work the authors have put into contextualising their observations and measurements, drawing together data from the literature in a systematic way, has really paid off. I believe this paper will be a valuable (and very citable) resource for future workers seeking to understand the configuration of the muscles of the mammalian face.

---

## Round 0.3 · accepted · Accept

Thank you for your attention to the two minor points raised in the second round of review. I'm happy to recommend your manuscript for publication, and look forward to seeing the paper published.